# A Machine Learning-Based Online Prediction Tool for Predicting Short-Term Postoperative Outcomes Following Spinal Tumor Resections

**DOI:** 10.3390/cancers15030812

**Published:** 2023-01-28

**Authors:** Mert Karabacak, Konstantinos Margetis

**Affiliations:** Department of Neurosurgery, Mount Sinai Health System, New York, NY 10029, USA

**Keywords:** spine surgery, spinal tumors, artificial intelligence, machine learning, NSQIP, prediction, online prediction tool

## Abstract

**Simple Summary:**

The overall incidence of spinal tumors in the United States was estimated to be 0.62 per 100,000 people. Surgical resection of spinal tumors intends to improve functional status, reduce pain, and, in some patients with isolated metastases or primary tumors, increase survival. Machine learning algorithms show great promise for predicting short-term postoperative outcomes in spinal tumor surgery. With this study, we aim to develop machine learning algorithms for predicting short-term postoperative outcomes and implement these models in an open-source web application.

**Abstract:**

***Background:*** Preoperative prediction of short-term postoperative outcomes in spinal tumor patients can lead to more precise patient care plans that reduce the likelihood of negative outcomes. With this study, we aimed to develop machine learning algorithms for predicting short-term postoperative outcomes and implement these models in an open-source web application. ***Methods:*** Patients who underwent surgical resection of spinal tumors were identified using the American College of Surgeons, National Surgical Quality Improvement Program. Three outcomes were predicted: prolonged length of stay (LOS), nonhome discharges, and major complications. Four machine learning algorithms were developed and integrated into an open access web application to predict these outcomes. ***Results:*** A total of 3073 patients that underwent spinal tumor resection were included in the analysis. The most accurately predicted outcomes in terms of the area under the receiver operating characteristic curve (AUROC) was the prolonged LOS with a mean AUROC of 0.745 The most accurately predicting algorithm in terms of AUROC was random forest, with a mean AUROC of 0.743. An open access web application was developed for getting predictions for individual patients based on their characteristics and this web application can be accessed here: huggingface.co/spaces/MSHS-Neurosurgery-Research/NSQIP-ST. ***Conclusion:*** Machine learning approaches carry significant potential for the purpose of predicting postoperative outcomes following spinal tumor resections. Development of predictive models as clinically useful decision-making tools may considerably enhance risk assessment and prognosis as the amount of data in spinal tumor surgery continues to rise.

## 1. Introduction

The overall incidence of spinal tumors in the US was estimated to be 0.62 per 100,000 people [1,2]. The majority of spinal tumors (up to 70%) are metastatic tumors. According to their location, spinal tumors are further divided into extradural (55%), intradural extramedullary (40%), and intramedullary (5%) [3,4]. Surgical resection of spinal tumors intends to improve functional status, reduce pain, and, in some patients with isolated metastases or primary tumors, increase survival [5,6,7]. Similar to other patients undergoing spine surgery, there is growing interest in finding the most effective ways to lower postoperative complications, length of hospital stays, and rate of nonhome discharges in the population of patients with spinal tumors [8,9]. Postoperative complications have a negative effect on a patient’s short-term quality of life, can lengthen their hospital stay, and can raise the cost of their medical care [10,11]. Several preoperative risk factors, such as preoperative functional status, disseminated malignancy, and poor baseline health, have been shown to predict higher complications and length of stay (LOS) [12,13].

In order to track and determine risk-adjusted estimates for these outcomes, emphasis is being placed on registries and databases as part of growing efforts to bend the healthcare cost curve. As a result, clinicians nowadays must manage vast amounts of complex data, which necessitates the employment of strong analytical techniques [14]. Machine learning (ML) algorithms can utilize high-dimensional clinical data to create precise patient risk assessment models, contribute to the formation of smart guidelines, and influence healthcare decisions by tailoring care to patient needs. In comparison to traditional prognostic models, which usually incorporate logistic regression, ML provides significant advantages. First, ML hardly ever requires prior knowledge of key predictors [15]. Second, compared to logistic regression, ML often has fewer restrictions on the number of predictors used for a given dataset. In large datasets with a considerable number of predictors, ML is useful since associations between predictors and outcomes may not always be instantly evident. Third, complex, nonlinear correlations in datasets that are more challenging to express and interpret using logistic regression can be discovered through ML [16]. These benefits often lead to ML being more accurate and robust than logistic regression techniques on the same dataset [17,18].

Based on our literature search, no study has explored the ability of ML algorithms to predict prolonged LOS, nonhome discharges, and postoperative complications in a single study following surgery for spinal tumors, without dividing into subtypes. This study aimed to assess the efficacy of machine learning algorithms in predicting postoperative outcomes after spinal tumor resection and create a user-friendly and accessible predictive tool for this purpose.

## 2. Materials and Methods

### 2.1. Data Source

Data for this study is from the American College of Surgeons (ACS) National Surgical Quality Improvement Program (NSQIP) database, which was queried to identify spinal tumor patients who were surgically treated from 2016 to 2020. We chose the most recent five years of data to take into account the advances in medicine. The ACS-NSQIP database is a national surgical registry with over 700 participating medical centers across the US for adult patients who underwent major surgical procedures across all subspecialties, except for trauma and transplant [19,20]. The data for each case, including demographics, preoperative comorbidities, operative variables, and 30-day postoperative outcomes, are being gathered by trained, skilled clinical reviewers [21]. Regular database auditing guarantees high-quality data with a previously reported interobserver disagreement rate of less than 2% in 2020 [22]. Detailed information about the database and data collection methods have been provided elsewhere [23].

### 2.2. Guidelines

We followed Transparent Reporting of Multivariable Prediction Models for Individual Prognosis or Diagnosis (TRIPOD) [24] and Journal of Medical Internet Research (JMIR) Guidelines for Developing and Reporting Machine Learning Predictive Models in Biomedical Research [25]. This was a retrospective machine learning classification study (outcomes were binary categorical) for prognostication in spinal tumors.

### 2.3. Study Population

We queried the NSQIP database to identify patients in whom the following inclusion criteria were met: (1) elective surgery, (2) inpatient operation, (3) current procedural terminology (CPT) codes for surgical resection of spinal tumors, (4) operation under general anesthesia, (5) surgical subspecialty neurosurgery or orthopedics. CPT codes we used to define our cohort are provided in Appendix A. We excluded patients with the following criteria: (1) emergency surgery, (2) patients with preoperative ventilator dependence, (3) patients with any unclean wounds (defined by wound classes 2 to 4), (4) patients with sepsis/shock/systemic inflammatory response syndrome 48 h before surgery, (5) patients with ASA physical status classification score of 4 and 5 or non-assigned.

### 2.4. Predictor Variables

Predictor variables included variables within the NSQIP database that were deemed to be known prior to the occurrence of the outcome of interest. These included (1) demographic information: age, sex, race/ethnicity, BMI (calculated from the height and weight), transfer status; (2) comorbidities and disease burden: diabetes mellitus, current smoker within one year, dyspnea, history of severe chronic obstructive pulmonary disease (COPD), ascites within 30 days prior to surgery, congestive heart failure within 30 days prior to surgery, hypertension requiring medication, acute renal failure, currently requiring or on dialysis, disseminated cancer, steroid or immunosuppressant for a chronic condition, >10% loss of body weight in last 6 months, bleeding disorders, preoperative transfusion of ≥1 unit of whole/packed RBCs within 72 h prior to surgery, ASA classification, functional status prior to surgery; (3) preoperative laboratory values: serum sodium, blood urea nitrogen (BUN), serum creatinine, serum albumin, total bilirubin, serum glutamic-oxaloacetic transaminase (SGOT), alkaline phosphatase, white blood cell (WBC) count, hematocrit, platelet count, partial thromboplastin time (PTT), International Normalized Ratio of prothrombin time (PT) values, PT; (4) operative variables: surgical specialty, days from hospital admission to operation, CPT code for the procedure; (5) spinal tumor variables: tumor location (extradural, intradural). Definitions of these predictor variables are provided in the ACS-NSQIP PUF User Guides (https://www.facs.org/quality-programs/data-and-registries/acs-nsqip/participant-use-data-file/, accessed on 1 January 2023). For transfer status, the variable values other than ‘Not transferred (admitted from home)’ were grouped as ‘Transferred’; for diabetes, the variable values’ Non-Insulin’ and ‘Insulin’ were grouped as ‘Yes’; for dyspnea, the variable values ‘Moderate Exertion’ and ‘At rest’ were grouped as ‘Yes’. Race and ethnicity variables were aggregated into one column, ‘Race’. If the patients’ Hispanic ethnicity values were ‘Yes’, their ‘Race’ values were assigned as ‘Hispanic’ regardless of their original values.

### 2.5. Outcome of Interest

The primary outcomes were prolonged length of stay, which we defined as total length of stay greater than 75% of the included patient population, nonhome discharges, and major complications. We defined nonhome discharge by dichotomizing the variable discharge destination. If patients required additional levels of care upon discharge, a nonhome discharge destination was identified and included ‘Rehab’, ‘Skilled Care, Not Home’, and ‘Separate Acute Care’. Patients with unknown discharge destinations, hospice discharges, discharges to unskilled facilities, and patients who expired were not included. We defined major complications, based on the previous literature [26,27,28], as having one of these events post-operatively: deep incisional surgical site infection (SSI), organ/space SSI, wound disruption, unplanned reintubation, pulmonary embolism, being on a ventilator for more than 48 h, renal insufficiency, acute renal failure needing dialysis, cardiac arrest, myocardial infarction, bleeding requiring blood transfusions, deep vein thrombosis, sepsis, and septic shock. We did not include complications involving less serious events to major complications, such as superficial wound infection, pneumonia, and urinary tract infection.

### 2.6. Data Preprocessing

In order not to introduce bias with the exclusion of patients with missing values, we utilized imputation. Fifteen continuous variables contained at least one missing value. After excluding variables with missing values for more than 25% of the patient population, missing values for continuous variables were imputed using the nearest neighbor (NN) imputation algorithm [29]. A value generated from cases in the entire dataset is used to replace each missing value for cases with missing values using NN imputation algorithms [30]. The only categorical variable that contained missing values was the variable race, and its missing values were imputed as ‘Unknown’.

The robust scaler was utilized to scale continuous variables to account for outliers [31]. Additionally, normalization is essential for ensuring that all feature values are on the same scale and assigned the same weight. Each continuous variable (e.g., BMI, laboratory values) was put on the (0, 1) range using a min–max normalization [32]. Categorical nonbinary variables (e.g., race, CPT codes) were one-hot-encoded, and variables with ordinal characteristics (e.g., ASA classification, functional status) were coded with the ordinal encoder [33].

The adaptive synthetic sampling (ADASYN) approach for imbalanced learning was used to artificially generate cases of positive outcomes of interest (i.e., prolonged LOS, nonhome discharges, major postoperative complications) based on the training and validation sets in order to overcome the class imbalance for a positive outcome of interest [34]. In order to enhance model learning and generalizability, ADASYN uses instances from the minority class that are difficult to learn and creates synthetic new cases based on these instances [35].

### 2.7. Training, Validation, and Test Sets

Data was split into training, validation, and test sets. The training set was used to develop the models, the validation set to adjust hyperparameters, and the test set to assess model performance. Data from 2015 to 2020 was split into training, validation, and test sets in a 60:20:20 ratio.

### 2.8. Modeling

Four supervised ML algorithms were utilized using the predictor variables to predict the outcomes: XGBoost, LightGBM, CatBoost, and random forest. We used the Optuna optimization library, where the optimized metric was the area under the receiver operating characteristic curve (AUROC). Optuna is a software framework for hyperparameter optimization that makes it simple to apply various state-of-the-art optimization techniques to carry out hyperparameter optimization quickly and effectively. To generate AUROC estimates that would serve as a guide for the optimization process, Tree-Structured Parzen Estimator Sampler (TPESampler) was employed as the Bayesian optimization algorithm. The final models for the outcomes were then built using the whole training set along with the optimized hyperparameters. ML analyses were performed in Python version 3.7.15.

### 2.9. Performance Evaluation

Models were evaluated graphically with receiver operating characteristic (ROC) curve, precision–recall curve (PRC), and calibration plots; and numerically with AUROC, area under PRC (AUPRC), accuracy, precision, recall, and Matthew’s correlation coefficient (MCC).

The ability of a binary classifier system to discriminate between positive and negative cases is shown graphically in a ROC curve, and the AUROC summarizes the model’s ability to do so. An AUROC of 1.0 indicates a perfect discriminator, whereas values of 0.90 to 0.99 are regarded as excellent, 0.80 to 0.89 as good, 0.70 to 0.79 as fair, and 0.51 to 0.69 as poor [36].

The model’s ability to detect all positive cases without recognizing false positives is shown graphically in a PRC, which plots recall (sensitivity) against precision (positive predictive value) and is summarized by the AUPRC. AUPRC can be a more responsive metric when used with datasets where the positive class is relatively uncommon because PRCs assess the proportion of correct predictions among the positive predictions [37].

In addition to the performance plots and metrics, we also utilized Shapley additive explanations (SHAP) to investigate the relative importance of predictor variables. SHAP is a visualization method frequently used in ML to comprehend how models make predictions.

### 2.10. Online Prediction Tool

We created a web application for getting predictions for individual patients based on their characteristics (Figure 1). This application is based on the models presented in this study with a few differences in implementation. The application and its source code are accessible on a platform that allows users to share ML models, Hugging Face (https://huggingface.co/spaces/MSHS-Neurosurgery-Research/NSQIP-ST, accessed on 1 January 2023).

### 2.11. Statistical Analysis

The descriptive analyses were reported as means (±standard deviations) for normally distributed continuous variables, medians (interquartile ranges) for non-normally distributed continuous variables, and number of patients (% percentages). Group differences in outcomes were tested with the independent t-test for normally distributed continuous variables with equal variances, the Welch’s t-test for normally distributed continuous variables with unequal variances, the Mann–Whitney U test for non-normally distributed continuous variables, and the Pearson’s chi-squared test for categorical variables. Normality was evaluated with the Shapiro–Wilk test, and Levene’s test was used to assess the equality of variances for a variable. The differences were considered to be statistically significant at *p* < 0.05. All statistical analyses were performed in Python version 3.7.15.

## 3. Results

Initially, a total of 6060 patients were identified via CPT codes. Inclusion and exclusion criteria were applied in a sequential manner. A total of 2449 were excluded due to non-elective surgeries, 145 due to outpatient surgeries, 17 due to anesthesia techniques other than general anesthesia, 42 due to surgical specialties other than neurosurgery or orthopedic surgery, 5 due to emergency surgeries, 8 due to preoperative ventilator dependency, 55 due to unclean wounds, 67 due to preoperative SIRS or sepsis, 165 due to ASA class 4, 5 or none assigned, 20 due to unknown LOS, 3 due to unknown major complication status and 11 due to discharge destination (Figure 2). After exclusion, 3073 patients were left in the analysis. There were 752 patients with prolonged LOS, 718 with nonhome discharges, and 379 with major complications. Characteristics of the patient population, both among the groups and in total, are presented in Appendix A.

The most accurately predicted outcomes in terms of AUROC and accuracy were the prolonged LOS with a mean AUROC of 0.745 and accuracy of 0.804, and the major complications with a mean AUROC of 0.730 and accuracy of 0.856. The most accurately predicting algorithm in terms of AUROC was random forest, with a mean AUROC of 0.743, followed by LightGBM, with a mean AUROC of 0.729. The mean AUROCs for CatBoost and XGBoost were 0.726 and 0.704, respectively. Detailed metrics regarding the algorithms’ performances are presented in Table 1. AUROC and AUPRC curves for the three outcomes are shown in Figure 3 and Figure 4.

SHAP plot of the XGBoost model for the outcome prolonged LOS, the CatBoost model for the outcome nonhome discharges, and the random forest model for the outcome major complications are presented in Figure 5. The other SHAP plots can be seen in Appendix A.

## 4. Discussion

This study presents a set of ML algorithms that can preoperatively predict prolonged LOS, nonhome discharges, and major complications for patients undergoing spinal tumor resection. The results of the study here demonstrate significant potential for the prediction of surgical outcomes and may help in the risk stratification process for spinal tumor resections. Patients who are at risk of unfavorable outcomes after spinal tumor resection can be better informed about the risks of surgery, and providers can better customize patient care plans to reduce the risk of these unfavorable outcomes. This paper contributes to the literature by demonstrating the efficacy and significance of incorporating machine learning into clinical settings to predict postoperative spine surgery outcomes [38].

The ML algorithms were able to predict between 78.9% and 81.1% of the patients who had prolonged LOS accurately with AUROC values between 0.726 and 0.760; between 72.8% and 76.4% of the patients who had nonhome discharges accurately with AUROC values between 0.650 and 0.725, and between 85.2% and 86.2% of the patients who had major complications accurately with AUROC values between 0.718 and 0.749 in the test set. Based on prediction mean accuracies and AUROC values for the different outcomes, the random forest algorithm was found to have performed the best among all the algorithms tested. These results can be deemed as fair classification performance, as previously explained.

In addition to reporting our methods and results in this paper, we created a web application accessible to physicians worldwide. Our web application not only allows users to see predictions for the three investigated outcomes by the four different algorithms utilized in this study, but it also allows them to see visual explanations of the predictions by SHAP values and figures. This additional interpretability might be of benefit in clinical settings, where providers can address the individual risk factors to achieve the best possible postoperative outcomes for their patients. To the best of our knowledge, this is the first available ML-based web application that enables users to have predictions with explanations for postoperative outcomes after spinal tumor resections.

Yang et al. posted an online calculator based on their regression-based nomogram for spinal cord astrocytomas using patient data in the Surveillance, Epidemiology, and End Results (SEER) Program of the National Cancer Institute [39]. First of all, the patients included in this study were diagnosed between 1975 and 2016. It is a very broad timespan and it is not reported how the year of diagnosis has impact on the individual survival predictions because the online calculator does not have an input for the year of diagnosis. Despite achieving comparable results with our study in terms of classification performance, this online calculator does not incorporate advanced analytical techniques, such as the ML algorithms we utilized in our study. Our web application provides predictions by four ML algorithms, which allows users to have multiple insights for a single patient. Moreover, the input for the mentioned tool includes variables like ‘histologic type’, ‘WHO grade’ and ‘postoperation radiotherapy’ which would not be known prior to surgery. This approach would not make personalized treatment plans possible preoperatively. With this tool, users can have overall and cancer-specific survival predictions, while users can have predictions for 30-day postoperative outcomes with our web application. Previously, Karhade et al. incorporated only the best ML algorithm across the model performance metrics for predicting 30-day mortality after surgery for spinal metastases into an interactive interface web application [40]. This application only allows seven variables as input and predicts one outcome, thirty-day mortality. Although Karhade et al. used ML algorithms as classifiers, they did not mention the details of implementation in the paper. Preprocessing of the data was not mentioned except the imputation of missing data and, the source code for the classification algorithms and online application were not shared. The above factors limit the reproducibility of the results. Moreover, both of these tools lack elaboration of the predictions, unlike our application which provides that via SHAP values and figures.

The measures presented here for the machine learning algorithms are congruent with the current literature. The outcomes we picked for this investigation have not been studied in a single study using ML algorithms; nevertheless, a few publications have investigated the classification performance of machine learning algorithms in predicting postoperative outcomes in spinal tumor patients employing different data sources. Using institutional data, Masaad et al. compared the performance of the metastatic spinal tumor frailty index (MSTFI) with ML methods in identifying measures of frailty as predictors of outcomes [41]. The random forest algorithm performed best in the study and had an AUROC value of 0.62 for postoperative complications. Jin et al. queried IBM MarketScan Claims Database for adult patients receiving surgery for intradural tumors, with their primary outcomes of interest being nonhome discharges and 90-day post-discharge admissions [42]. Their classification models were developed using a logistic regression approach regularized by the least absolute shrinkage and selection operator (LASSO) penalty, and they obtained AUROC values of 0.786 for nonhome discharges and 0.693 for 90-day readmissions.

Although a few studies using the NSQIP database that analyzed the accuracy of machine learning algorithms in predicting postoperative outcomes included, we did not include some of the available variables that would not be known prior to the surgery as input to our models, like total operative time [43,44]. The length of the procedure may be a consequence of unfavorable outcomes rather than its cause [45]. The study, in which Kalagara et al. analyzed the NSQIP database for readmissions following lumbar laminectomy and developed predictive models to identify readmitted patients, reported an overall accuracy of 95.9% and an AUROC value of 0.806 with a gradient boosting machine (GBM) model using all patient variables [46]. The most important variables that made this model achieve such good results included post-discharge complications and discharge destinations. A second GBM model to predict readmission utilizing only information known prior to readmission had an accuracy of 79.6% and an AUROC of 0.690. Still, this model included postoperative variables such as discharge destination and total hospital LOS, and those were among the most important features.

The SHAP analysis results are in line with the current literature on regression analysis for the relative importance of predictor variables. A study on predictors of discharge disposition following laminectomy for intradural extramedullary spinal tumors identified age over 65 years, ASA classification over three, and dependent functional status as predictors of nonhome discharge [13]. These variables were among the most important features of our machine learning models, as can be seen from the SHAP plots.

The study does have some potential limitations despite the strength of the methodology described. First, the sample of patients undergoing spinal tumor resection may not have accurately represented all the patients who undergo spinal tumor resection. The NSQIP dataset depends on reporting from participating hospitals. As a result, patients from hospitals with the infrastructure to uphold NSQIP reporting requirements will be overrepresented in the sample of spine tumor patients between 2016 and 2020. In addition, coding errors and other inaccuracies always affect studies using an extensive clinical database. Even though the NSQIP database is frequently used, there have been a few studies evaluating its actual accuracy. Neurosurgical procedure CPT codes contain numerous internal inconsistencies, according to Rolston et al. [47]. Second, NSQIP data do not include specific factors that might be associated with a patient’s risk of unfavorable postoperative outcomes. For example, we could not assess the effect on outcomes of tumor-specific variables, such as histologic type or tumor size, because we lacked access to more granular data. While the current mean AUROCs between 0.703 and 0.734 are fair, adding these and other relevant variables may enhance the algorithm’s performance.

## 5. Conclusions

Machine learning algorithms show great promise for predicting postoperative outcomes in spinal tumor surgery. These algorithms can be incorporated into clinically practical decision-making tools. The development of predictive models and the use of these models as accessible tools may significantly improve risk management and prognosis. Herein, we present and make publicly available a predictive algorithm for spinal tumor surgery aiming at the above goals.

## Figures and Tables

**Figure 1 cancers-15-00812-f001:**
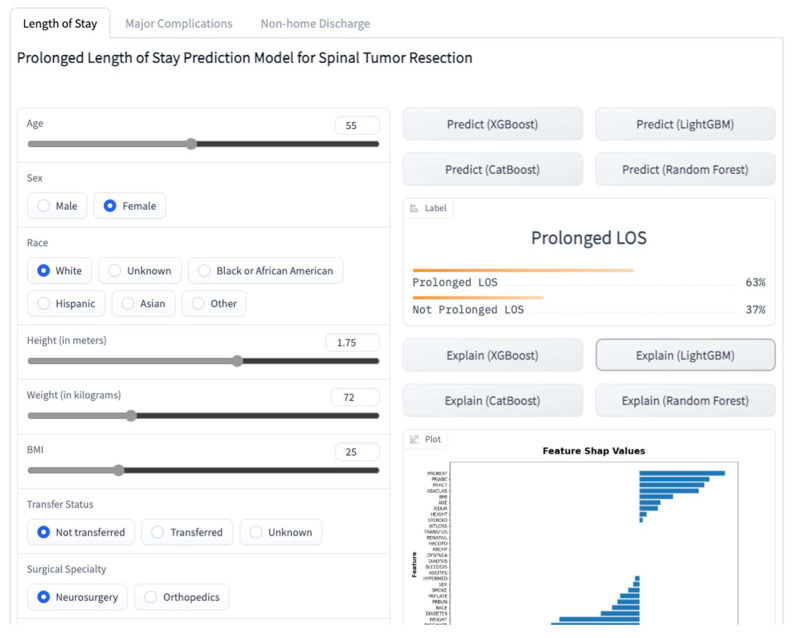
A screenshot of the online web application.

**Figure 2 cancers-15-00812-f002:**
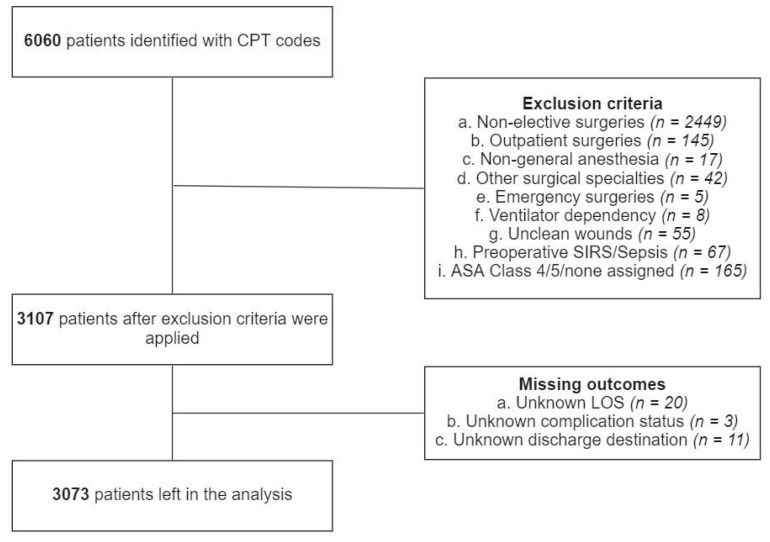
Patient selection process.

**Figure 3 cancers-15-00812-f003:**
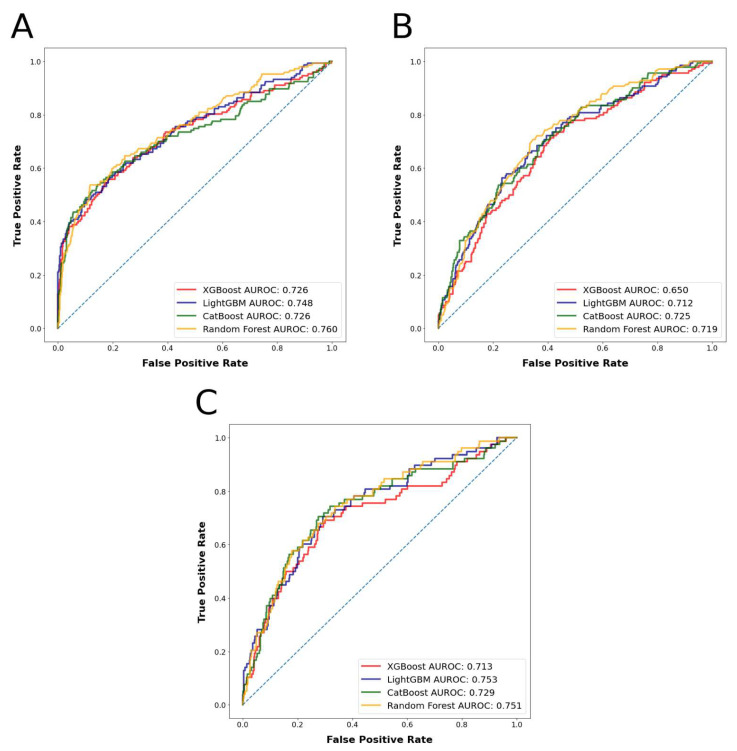
(**A**): Algorithms’ receiver operator curves for the outcome prolonged length of stay. (**B**): Algorithms’ receiver operator curves for the outcome nonhome discharges. (**C**): Algorithms’ receiver operator curves for the outcome major complications.

**Figure 4 cancers-15-00812-f004:**
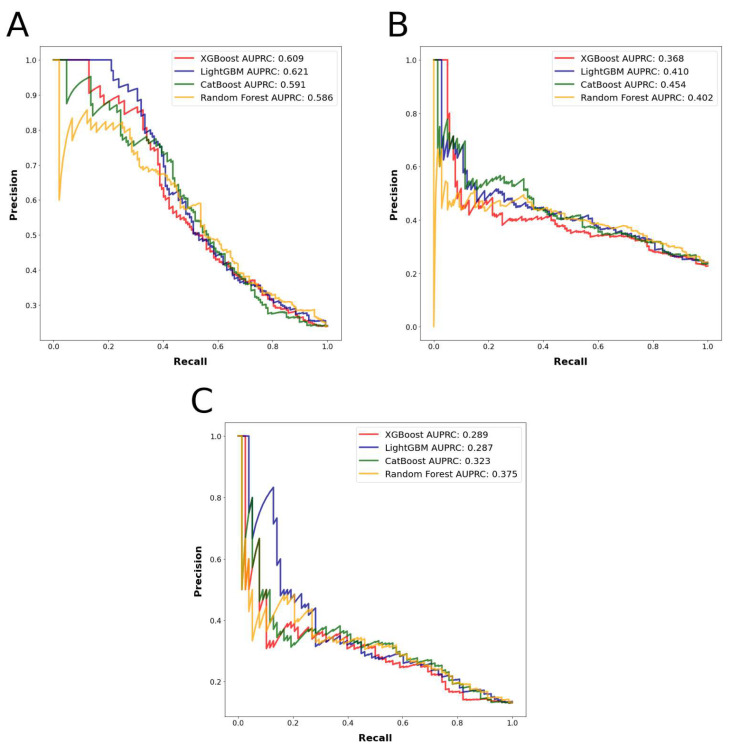
(**A**): Algorithms’ precision–recall curves for the outcome prolonged length of stay. (**B**): Algorithms’ precision–recall curves for the outcome nonhome discharges. (**C**): Algorithms’ precision–recall curves for the outcome major complications.

**Figure 5 cancers-15-00812-f005:**
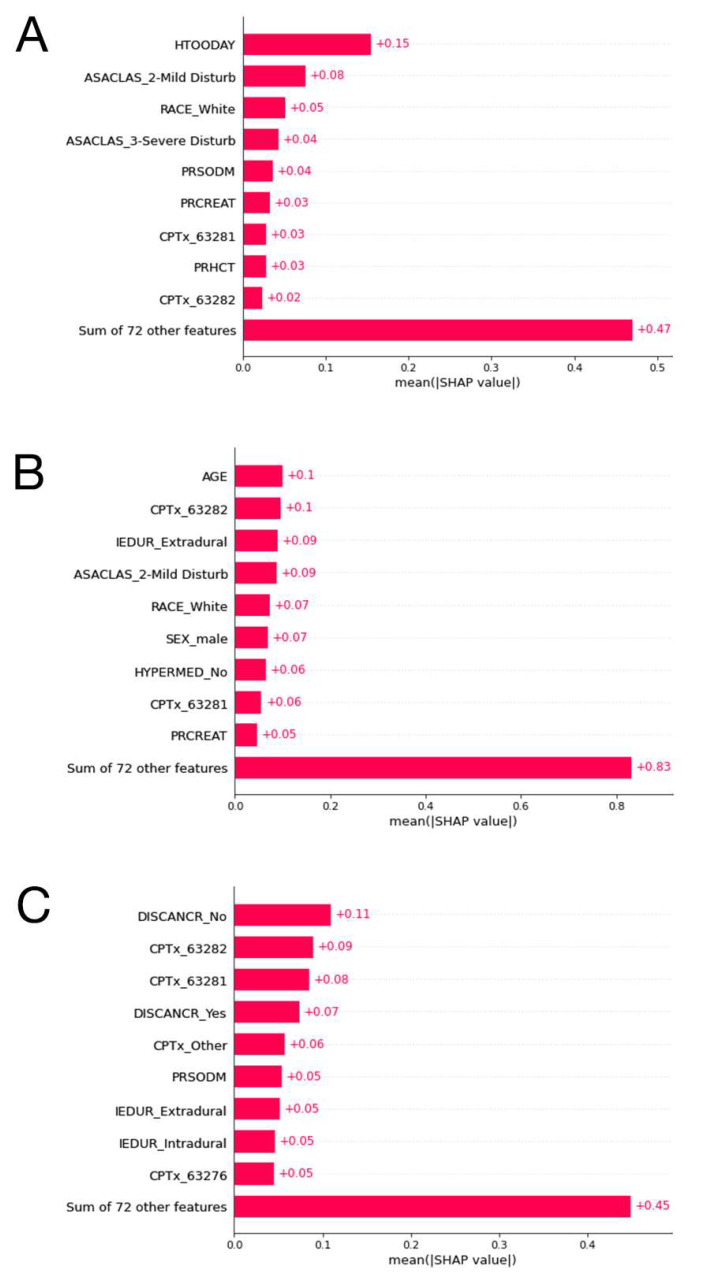
(**A**): The ten most important features and their mean SHAP values for the model predicting prolonged length of stay with the LightGBM algorithm. (**B**): The ten most important features and their mean SHAP values for the model predicting nonhome discharges with the CatBoost algorithm. (**C**): The ten most important features and their mean SHAP values for the model predicting major complications stay with the random forest algorithm.

**Table 1 cancers-15-00812-t001:** Metrics regarding the algorithms’ performances.

Outcome	Algorithm	P	R	F1	MCC	AUPRC	ACC	AUROC
**LOS**	XGB	0.503	0.565	0.532	0.398	0.609	0.789	0.744
	LGB	0.449	0.641	0.528	0.423	0.621	0.808	0.748
	CB	0.469	0.645	0.543	0.437	0.591	0.811	0.726
	RF	0.490	0.621	0.548	0.431	0.586	0.807	0.760
	**Mean**	**0.478**	**0.618**	**0.538**	**0.422**	**0.602**	**0.804**	**0.745**
**NHD**	XGB	0.307	0.381	0.340	0.173	0.368	0.728	0.650
	LGB	0.343	0.475	0.398	0.262	0.410	0.764	0.712
	CB	0.436	0.477	0.455	0.304	0.454	0.763	0.725
	RF	0.414	0.436	0.425	0.261	0.402	0.745	0.719
	**Mean**	**0.375**	**0.442**	**0.405**	**0.250**	**0.408**	**0.750**	**0.701**
**MC**	XGB	0.192	0.405	0.261	0.212	0.293	0.862	0.718
	LGB	0.192	0.375	0.254	0.197	0.305	0.857	0.726
	CB	0.244	0.373	0.295	0.222	0.321	0.852	0.728
	RF	0.256	0.377	0.305	0.231	0.318	0.852	0.749
	**Mean**	**0.221**	**0.383**	**0.279**	**0.216**	**0.309**	**0.856**	**0.730**

P, precision; R, recall; MCC, Matthew’s correlation coefficient; AUPRC, area under the precision recall curve; ACC, accuracy; AUROC, area under the receiver operating characteristic curve; LOS, length of stay; NHD, non-home discharge; MC major complications; XGB, XGBoost; LGB, LightGBM; CB, CatBoost; RF, Random Forest.

## Data Availability

Restrictions apply to the availability of these data. Data were obtained from American College of Surgeons National Surgical Quality Improvement Program and are available https://www.facs.org/quality-programs/data-and-registries/acs-nsqip/ (accessed on 1 January 2023) with the permission of American College of Surgeons.

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
