# Peer review of "A Machine Learning-Based Online Prediction Tool for Predicting Short-Term Postoperative Outcomes Following Spinal Tumor Resections"

_cancers, 2023, doi:10.3390/cancers15030812_

Round 1

Reviewer 1 Report

The article provides a useful decision-making tool of pre-operatively predicting short term postoperative outcomes following spinal tumor operation via machine learning. It makes sense in clinical.  However, the research design should be improved. Firstly, the article enrolled the benign, malignant and metastatic spinal tumors. Among which. the focus of the latter two kinds of spinal tumors was how to improve the survival, rather than lowing the postoperative complications. Secondly, the postoperative outcomes varied from the location of spinal cord, such as extradural, intradural extramedullary and intramedullary locations which maybe lead to the inconsensus result. Thirdly, some potential factors affecting the postoperative outcomes had not evaluated in the article such as the pre-operative radiologic characteristics with/without capsular, with/without hemorrhage, histological pathology, Ki67 and so on.

Author Response

Thank you very much for reviewing our paper.

  1. The purpose of the current study was to explore the ability of machine learning algorithms to predict prolonged LOS, nonhome discharges, and postoperative complications in a single study following surgery for spinal tumors, without dividing into subtypes. That was why we did not separate spinal tumors into subcategories.
  2. The models proposed in the study take the location of tumors (intradural versus extradural) into account when making predictions. Please see our web application for implementation (https://huggingface.co/spaces/MSHS-Neurosurgery-Research/NSQIP-ST).
  3. Third, we would love to have cancer-specific variables such as the pre-operative radiologic characteristics with/without capsular, with/without hemorrhage, histological pathology, Ki67, but the NSQIP database we utilized in our study does not have these variables. We will explore more cancer-specific variables and outcomes in future studies, with more granular datasets.

Reviewer 2 Report

The authors should be commended for their interesting study on this important topic.

Please find following my comments:

1) Methods: "We defined major complications, based on the previous literature, as having one of these events post-operatively [..]" Please cite the relevant literature you are referring to.

2) Results: Please include a figure/diagram describing the inclusion/exclusion process with the relevant numbers at each stage.

Author Response

Thank you very much for reviewing our paper.

  1. Citation added.
  2. Figure 2 has been added.